# *MingOfficial*: A Ming Official Career Dataset and a Historical Context-Aware Representation Learning Framework

**You-Jun Chen[1*], Hsin-Yi Hsieh[1†], Yu-Tung Lin[1†], Yingtao Tian[2§], Bert Wang-Chak Chan[2§],**
**Yu-Sin Liu[3†], Yi-Hsuan Lin[3†], and Richard Tzong-Han Tsai[1‡†,C],**

[†]Department of Computer Science and Information Engineering, National Central University, Taoyuan, Taiwan

[*]Courant Institute of Mathematical Sciences, New York University, NY, USA

[‡]Center for GIS, RCHSS, Academia Sinica, [§]Google DeepMind

*yc7093@nyu.edu, {hsinmosyi, judy502203, yusinliu, julialin, thtsai}@g.ncu.edu.tw, {alantian, bertchan}@google.com*

## Abstract

In Chinese studies, understanding the nuanced traits of historical figures, often not explicitly evident in biographical data, has been a key interest. However, identifying these traits can be challenging due to the need for domain expertise, specialist knowledge, and context-specific insights, making the eprocess time-consuming and difficult to scale. Our focus on studying officials from China's Ming Dynasty is no exception. To tackle this challenge, we propose *MingOfficial*, a large-scale multi-modal dataset consisting of both structured (career records, annotated personnel types) and text (historical texts) data for $13,031$ officials. We further couple the dataset with a graph neural network (GNN) to combine both modalities in order to allow investigation of social structures and provide features to boost down-stream tasks. Experiments show that our proposed *MingOfficial* could enable exploratory analysis of official identities, and also significantly boost performance in tasks such as identifying nuance identities (e.g. civil officials holding military power) from $24.6\%$ to $98.2\%$ $F_1$ score in hold-out test set. By making *MingOfficial* publicly available at https://data.depositar.io/en/dataset/ming_official as both a dataset and an interactive tool, we aim to stimulate further research into the role of social context and representation learning in identifying individual characteristics, and hope to provide inspiration for computational approaches in other fields beyond Chinese studies.

## 1 Introduction

Investigating political figures has been a key interest in Chinese historical research. Techniques such as prosopography, which systematically collates all relevant biographical data, have been employed to

---

The numbers following the author names indicate their order of authorship.

[C]Corresponding author. Email: thtsai@g.ncu.edu.tw

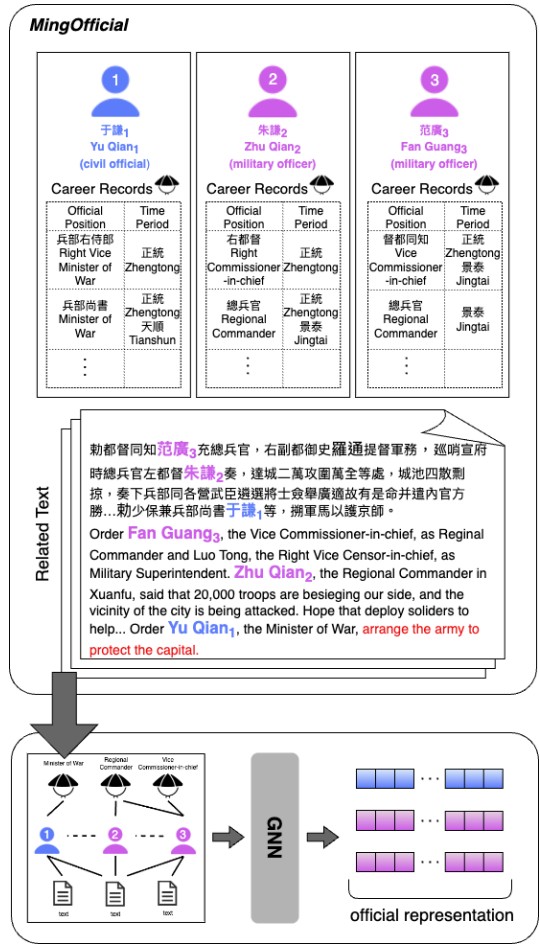

Figure 1: Overview of our approach: We start by proposing a dataset that combines *structured* career records and *textual* historical records of officials from the Ming Dynasty. These diverse sources of information are integrated using a Graph Neural Network (GNN)-based model to aggregate multi-modal data into unified embeddings. The learned representation captures the nuanced relations between officials, allowing high performance downstream tasks qualitatively and quantitatively in Ming history study.

examine shared histories of past groups or individuals (Stone, 1971; Verboven et al., 2007; Lunding et al., 2020). Recently with the trend of digital humanities, databases such as CBDB (Harvard Uni-

versity et al., 2019) have brought a transformative shift in this field, offering scholars and historians a more flexible approach to studying Chinese political figures (Tsui and Wang, 2020). These databases allows answering queries like "Who entered government through the civil examination from Jiangnan region during the Hongzhi reign (1487-1505)?" and have proven useful for studying officials in China's Ming Dynasty, which is our focus in Chinese study.

However, identifying nuanced characteristics not explicitly presented in biographical data remains challenging. For example, the possession of military power by civil officials — an official with a position in the War Ministry (兵部), a department overseeing military administration (Hucker, 1998), in the career record does not necessarily having military power unless textual record suggests military campaigns under the official's command. Identifying these cases often requires specialized knowledge and contextual understanding. Undoubtedly, such process is time-consuming.

To tackle this challenge, we hypothesize that alongside career records, an approach considering social relations among civil officials along with their notable activities described in records, would provide much needed insights. In doing so, we propose *MingOfficial*, a large-scale multi-modal dataset consisting of two modalities: (1) structured data, the career records of $13,031$ Ming dynasty officials, reflecting their political standing and social prestige, and (2) textural data, $69,688$ paragraphs from the records in *Ming Shi Lu* (明實錄, or *Imperial Annals of Ming Dynasty*), documenting major activities an official participated in. This data, organized into a graph structure, is coupled with graph neural networks (GNN) (Scarselli et al., 2009) to embed historical context and social dynamics into a lower-dimensional space, enhancing downstream tasks.

We empirically analyze the dataset and apply it to down-stream tasks. The dataset in its vanilla form already could reflect the changing power dynamics over time and shows the latent social relations among officials. Furthermore, our graph neural network model enhances the identification of nuanced traits, boosting the $F_1$ score from $24.6\%$ to $98.2\%$ in distinguishing civil officials with military power. This substantial improvement is achieved despite being trained on a mere $6\%$ of labeled data, composed of only 49 instances, with just 8 positive

cases. This demonstrates the practicality of our approach for tasks identifying character traits, which often require costly annotations entailing meticulous reading of scattered historical records about an individual.

The potential of *MingOfficial* extends beyond its historical implications. It serves as a comprehensive knowledge base for numerous research fields, including network analysis, sociological studies, computational linguistics, and natural language processing. It also provides valuable insights into the Ming dynasty's political and social dynamics, forming a strong foundation for comparative historical and political studies. Our contributions are the following:

- We are the first to use both career records and relevant textual content to represent past political figures.
- We introduce *MingOfficial*, a dataset that includes Ming dynasty officials' career records, position categories, annotated personnel types, and related historical texts, providing a deep understanding of the political network and governmental system of Eastern institutions.
- We propose to learn the representation of Ming officials using graph neural network, fully exploiting the rich, interconnected information in *MingOfficial*.
- We apply these representations to identify nuanced characteristics, using the military power of civil officials as an example, and offer annotation guidelines for future research. Our approach establishes a foundation for future research to further uncover intricate characteristics of past political figures.

## 2 The *MingOfficial* Dataset

In this section, we discuss the challenges of identifying nuanced characteristics of civil officials' military power, the sources and preprocessing methods for our dataset, and the process of reliability of annation in our dataset.

### 2.1 Nuanced Characteristics as a Challenge

Identifying characteristics of a historical figure can be challenging due to the often implicit nature of such information in biographical data. One example is the military power wielded by civil officials, as a substantial but not all of them constitutes is proficient in both civil and military affairs (Filipiak, 2012). Correctly identifying these individuals re-

quires deep domain knowledge and an extensive understanding of the historical backdrop of the Ming Dynasty.

For example, Yu Qian (于謙), a civil official whose career records are presented in Figure 1, held a significant position in the War Ministry ([Hucker, 1998]). Merely having such a role does not unequivocally imply military power, as the ministry mainly dealt with logistics, personnel, finances, and infrastructure rather than active commanding. However, historical accounts, as in Figure 1, provide context showing that Yu Qian was tasked with arranging the army to protect the capital during a war event, thus suggesting his military power.

## 2.2 Dataset Construction

The construction of dataset involves collection and processing from two kinds of data source: structured data and textual data, which we detail as follows.

**Structured Data** We gather personal information on 13,031 officials who served in the Ming Dynasty from a biographical database ([IHP]),including names, government entry modes, and raw career records. Given the variance in descriptions of identical positions, we organize these records into 123 categories according to *Da Ming Hui Dian* (大明會典, or *The Code of the Ming Dynasty*) and *Chinese-English Dictionary of Ming Government Official Titles* ([Zhang et al., 201]). This categorization reflects the administrative structure of the Ming government, grouping titles of core officials and subordinates from different departments into office categories, facilitating the conversion of career information into a graph structure. Examples of this categorization are available in Appendix A.

Each official's personnel type is inferred from the position categories in their career records, with five groups: civil officials, military officers, eunuchs, imperial family members, and others. Figure 2 (a) shows the personnel types distribution among all Ming officials, with civil officials forming the majority within the office system. Note that our method effectively handles such imbalanced labels, as demonstrated in Section 4.

**Textual Data** We source related texts from *Ming Shi Lu* (明實錄, or *Imperial Annals of Ming Dynasty*), a collection of thirteen historical books by Ming officials, available at [Academia Sinica (2019)]. These documents cover the reigns of the thirteen Ming Emperors, from the Hongwu Emperor (洪

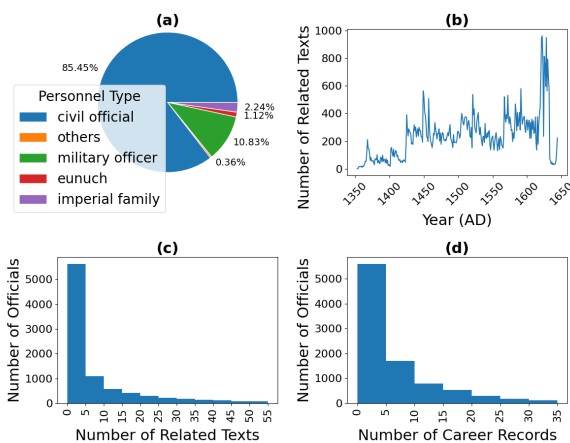

Figure 2: Statistics of the *MingOfficial* Dataset. (a) Distribution of personnel types. (b) Annual count of text entries. (c) Distribution of the number of related text entries per official. (d) Distribution of the number of career records per official.

武帝, 1368-1398) to the Chongzhen Emperor (崇禎帝, 1627-1644). They include imperial edicts, laws, and major political, economic, and cultural events, serving as a fundamental resource for Ming history research.

We first process the texts in *Ming Shi Lu* using named entity recognition ([Lai, 2016]) and named entity disambiguation ([Wu, 2021]) to identify people, location names, and official titles in the text. We then select only the paragraphs in which a person entity is linked to an official. As a result, we obtained a total of 69,688 processed texts. The number of processed texts each year is shown in Figure 2b, and a breakdown by each emperor is detailed in Appendix B.

## 2.3 The Process and Reliability of Annotation for Civil Officials' Military Power

Identifying military power among civil officials presents a unique challenge, requiring an understanding of both textual content and societal structure. The process we developed, grounded in historical context, could generalize to modern scenarios. It serves as a model for identifying specific characteristics in complex social networks.

We determine a civil official's military power based on evidence from primary historical sources. These sources include *Ming Shi Lu*, *Ming Shi* (明史, or *History of Ming*), and the Military System subdivisions of the Economic series in *Gujin Tushu Jicheng* (古今圖書集成, or *Imperial Encyclopaedia*). Officials are identified as having military power if they meet one or more of the following

Figure 3: Example of evidence indicating military power for the official Wang Shou Ren (王守仁).

| Annotators | AB | BC | AC |
|---|---|---|---|
| Cohen's Kappa | 0.80 | 0.80 | 0.72 |

Table 1: Inter-annotator agreement table, demonstrating substantial consensus among annotators.

criteria: (1) Personally leading soldiers to the battlefield, (2) Commanding troops and (3) Arranging the personnel in the military.

Should a source paragraph detail military engagements, be they wars or rebellions, we deem the associated official to hold military power, based on the given criteria. Consider Figure 3 as an illustration: it depicts Wang Shou Ren (also known as 王守仁 or 王陽明), a civilian authority, mobilizing troops to suppress a rebellion. This narrative provides clear evidence of his military prowess.

To assess the reliability of our annotations, we employ Cohen's kappa measure (Cohen, 1960) for inter-annotator agreement (IAA) (Artstein, 2017). This assessment involves 100 sampled civil officials from *MingOfficial*. Our annotators, proficient in literary Chinese and knowledgeable in history and humanities, included an author of the paper(A) and two individuals (B and C) not involved in criteria development. Table 1 shows their substantial agreement, validating the reliability of our approach.

## 3 Graph-based Representation Learning

An important part of our proposed method, depicted in Figure 4, involves graph-based representation learning. Concretely, we start with the construction of the graph and node initialization, using information in the *MingOfficial* dataset. We then apply Graph Neural Networks (GNN) (Scarselli et al., 2009) to learn the officials' representations. Finally, we utilize these learned representations to identify the military power of civil officials.

### 3.1 Graph Construction and Node Initialization

We construct a graph to represent Ming dynasty's complex political landscape, consisting of three node types and four kinds of relations. These nodes represent Ming officials (**P**), official position categories (**O**), and relevant historical texts (**T**). Among the relationships, **P-O**, **P-T**, **coP-P** depict an official and the office they held (political influence), an official and texts that mention it (historical events), and two officials concurrently cited within the same text, respectively (inter-official association).

Distinctively, **simP-P** is dynamically generated during the learning process of the GCNs. At each step, after updating the official embeddings with text embeddings using the **P-T** relation, officials with the top-$k$ Cosine similarity between their embeddings form the **simP-P** relationship, signifying similarities in their experiences as inferred from texts. Except for **coP-P**, all relationships are delineated by reign periods, marking the temporal aspect of an official's activity or relationships.

We initiate the *nodes* embedding as follow. For **T**, we use the CLS sentence embeddings generated by the pretrained BERT-based-Chinese model (Devlin et al., 2019), while for **P** and **O**, we use the metapath2vec model (Dong et al., 2017) that samples path to reflect the administrative structure of the Ming governmental institutions (see Figure 5).

### 3.2 Representation Learning

Officials' representations are learned using GNN which update node representations by aggregating information from each node's neighbors to update, effectively generating context-rich feature representation of the node. The learning includes two stages:

**Stage 1: Personnel Type Categorization**. The model's objective is to classify officials from the Ming Dynasty into three major political groups: civil officials, military officers, or eunuchs. It relies on analyzing their career records, with a central emphasis on **P-O**, encompassing all their held official positions. The model employs GNN to aggregate information from **O** to **P** through **P-O**. Following this, we feed the embedding of **P** (represented as $\mathbf{p_i}$) into a feed-forward neural network (FFNN) with softmax ($f$) to predict the personnel type $c_i$. Cross entropy loss is used:

$$L_{\mathrm{PT}} = - \sum_{i \in S_{\mathrm{ofc}}} \log f(c_i | \mathbf{p_i}) \qquad (1)$$

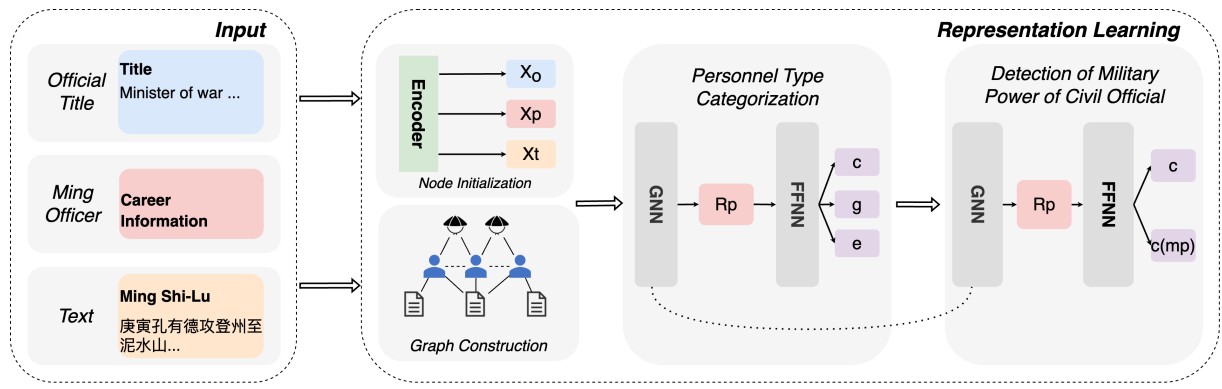

Figure 4: Our overall approach to detect the civil official's military power.

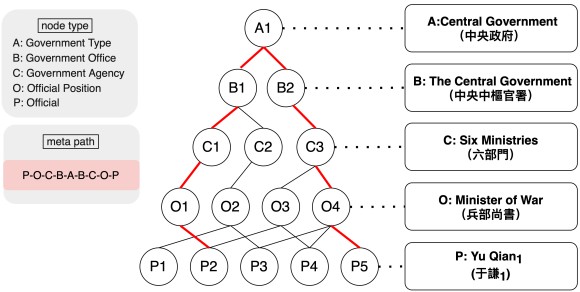

Figure 5: Our GNN based method samples paths for metapath2vec part, using the governmental organization structure in the Ming Dynasty. We designe the hierachy of nodes of type A, B, C, following the actual government hierarchy in Ming Dynasty.

$$f(c|\mathbf{p}) = \text{Softmax}(W_c\mathbf{p} + b_c) \qquad (2)$$

where $S_{\text{ofc}}$ is the set of officials for Stage 1, $W_c$ and $b_c$ are learnable parameter of the FFNN.

**Stage 2: Detection of Civil Officials with Military Power (MP).** Leveraging the parameters of the GNN model trained in the previous stage, the model refines the officials' representations **P** using **P-O** and **P-P** relations. We then feed the embedding of **P** ($\mathbf{p_i}$) into another FFNN with softmax ($f$) to predict the probability of **P** possesing military power ($m_i$). Cross entropy loss is used:

$$L_{\text{MP}} = -\sum_{i \in S_{\text{civ}}} \log f(m_i|\mathbf{p_i}) \qquad (3)$$

$$f(m|\mathbf{p}) = \text{Softmax}(W_m\mathbf{p} + b_m) \qquad (4)$$

where $S_{\text{civ}}$ is the set of civil officials for Stage 2, $W_m$ and $b_m$ are learnable parameter of the FFNN.

## 4 Experiments

In this section we start with our technical setup, and show Exploratory Analysis of *MingOfficial*, followed by the results and analysis of our method.

### 4.1 Technical Setup

**Models for Comparison** We compare our model with a non-GNN baseline of Multi-layer Perceptron (MLP) model and with GNN alternatives, including GraphSAGE Hamilton et al. (2018), GAT Veličković et al. (2018) , and HGT (Hu et al., 2020a) models using different combinations of relations. The specific structure of the MLP model, details regarding node initialization and model implementation can be found in Appendix F.

**Data Splits** We split our dataset into training, validation, and testing sets across two stages of representation learning (see 3.2) , as outlined in Table 2. In Stage 2, due to the rarity of civil officials with military power and the resource-intensive nature of annotation (see E, we have a significantly smaller dataset. For this stage, we apply 5-fold cross-validation and report the average and standard deviation of model performance across these five sets.

### 4.2 Exploratory Analysis of *MingOfficial*

We show that our *MingOfficial* dataset is valuable for examining historical socio-political dynamics, serving as a potent resource for exploring not only individual biography and prosopography, but also broader social dynamics such as class, mobility, and political networks, emphasizing its value for historical and sociological studies. More details regarding this analysis and relevant historical background can also be found in Appendix C.

First we display the changing prominence of civil officials, military officers, and eunuchs throughout the Ming Dynasty in Figure 6. Overall, this graph showcases the dynamic power shifts between the three major political groups over time. In particular, it provides a preliminary explana-

| Stage | Class | Train | Val | Test | Total |
|-------|-------|-------|-----|------|-------|
| Stage 1 | Civil Official (CO) | 4,381 (55.1%) | 1,096 (13.8%) | 1,370 (17.2%) | 6,847 (86.1%) |
| | Military Officer | 640 (8.1%) | 161 (2.0%) | 201 (2.5%) | 1002 (12.6%) |
| | Eunuch | 66 (0.8%) | 617 (0.2%) | 21 (0.3%) | 104 (1.3%) |
| Stage 2 | CO w/o MP | 41 (4.9%) | 28 (3.4%) | 622 (73.8%) | 691 (83.1%) |
| | CO w/ MP | 8 (1.0%) | 6 (0.7%) | 127 (15.3%) | 141 (16.9%) |

Table 2: Data Splits for Stage 1 and 2. For Stage 1, we sample 64% of the labeled data for training, 16% for validation, and 20% for testing. For Stage 2, we sample 6% for training, 4% for validation, and 90% for testing. The proportion of each class remains consistent across training, validation, and test sets. "CO w/o MP" stands for civil officials not having military power, while "CO w/ MP" stands for civil officials having military power.

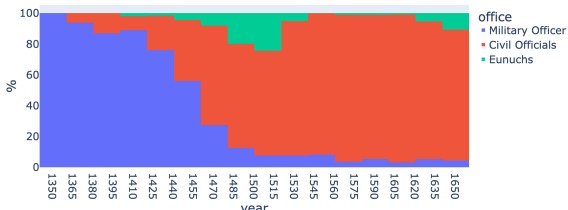

Figure 6: The official position power shift among three main groups influential in politics, over the Ming times.

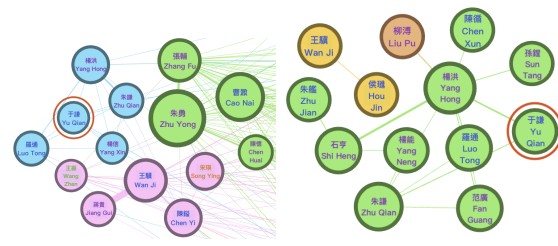

Figure 7: Co-occurrence network during Zhengtong era (AD 1449) on left and Jingtai era (AD 1450-1451) on right. Our figure of interest, Yu Qian, is circled in red.

| Model | Setting | Epoch | F1 (%) |
|-------|---------|-------|--------|
| **MLP** | P + O | 20 | $24.6_{4.2}$ |
| | P + O + T | | $\mathbf{28.8}_{5.4}$ |
| **SAGE** | P-O | 4 | $76.7_{2.5}$ |
| | P-O + coP-P | | $74.5_{2.1}$ |
| | P-O + simP-P | | $\mathbf{87.7}_{2.7}$ |
| | P-O + coP-P + simP-P | | $83.2_{2.7}$ |
| **HGT** | P-O | 4 | $58.2_{3.1}$ |
| | P-O + coP-P | | $83.0_{3.1}$ |
| | P-O + simP-P | | $\mathbf{84.5}_{4.0}$ |
| | P-O + coP-P + simP-P | | $74.3_{3.0}$ |
| **GAT** | P-O | 4 | $86.0_{1.6}$ |
| | P-O + coP-P | | $91.0_{1.5}$ |
| | P-O + simP-P | | $\mathbf{98.2}_{0.4}$ |
| | P-O + coP-P + simP-P | | $96.7_{0.8}$ |

Table 3: Model performance of detecting civil officials with military power under various settings. We report average and standard deviation (as subscripts) performance over 5 different data splits (see 4.1).

score for CO w/ MP in Table 3.

**Efficacy of Graph Structure with GNN** All GNN models consistently outperform the baseline MLP model by a significant margin. This notable improvement is evident despite the GNN models being trained for only 4 epochs, compared to the 20 epochs for the MLP model. These results underline the effectiveness of our graph-based approach in generating high-quality individual representations.

**Impact of Incorporting Textual Data** We found that integrating textual data, specifically official narratives, significantly enhances the model's performance. This trend is consistent across both the MLP and GNN models. For the MLP model, the use of text embeddings resulted in better performance than models without them. For GNN models, those incorporating coP-P or(and) simP-P along with P-O generally outperform those using P-O alone. The only exception to this trend was observed with SAGE, where P-O + coP-P per-

tion of the phenomena whereby military officers were heavily relied upon in the early Ming dynasty, and how the power in military discourse gradually shifted towards civil officials and eunuchs during the middle period.

Then we we construct co-occurrence networks based on the person mentions in texts, and show in Figure 7 the complex and changing social relations spanning across the reigns of two different emperors. The depicted variations in network structures between the two periods offer unique insights into the dynamics of interpersonal relationships and social structures over time.

### 4.3 Results

We evaluate the GNN and MLP models on the task of detecting civil officials with military power (CO w/ MP). We experiment with various settings for these models and present their performance as F1

forms worse than P-O alone. We attribute this to SAGE's lack of an attention mechanism which could weigh the importance of the relatively coarse co-occurrence relations. In contrast, GAT and HGT, with their attention mechanisms, can more effectively leverage coP-P.

**Comparison between coP-P and simP-P** Across all GNN models, the combination of P-O + simP-P consistently achieved the highest scores. Particularly, SAGE and GAT showed considerable improvements when using P-O + simP-P as opposed to P-O + coP-P (SAGE: +17.7%, GAT: + 8.0%). In contrast, HGT showed only a marginal increase. We also experimented with the incorporation of both coP-P and simP-P, which improved the performance of SAGE and GAT compared to P-O + coP-P. However, this setting negatively impacted the performance of HGT. Given that HGT treats the graph as a heterogeneous entity with different attention layers for various types of nodes, we hypothesize that it might require additional training time and data for model optimization.

**Best Performance and Training Efficiency** Notably, GAT achieves the most stable and highest F1 score of 98.2% for CO w/ MP under the setting of P-O + simP-P. This outstanding performance was achieved on a small training set of only 49 examples and with 4 training epochs, highlighting the efficiency of our approach. This finding is particularly important as it suggests the potential to significantly reduce annotation labor in future applications.

### 4.4 Error Analysis

We conducted an error analysis on the best performing GNN model – GAT, and present the confusion matrices for the prediction results under different settings in Figure 8. We glean several important insights as follows, with full details available in the Appendix G:

- Career records alone (P-O) may not be sufficient to identify nuanced characteristics, such as the military power among civil officials. In some cases, the model incorrectly predicted these characteristics based on seemingly similar career trajectories.
- Sparse individual information consistently caused false negatives across all settings. The need for enriched datasets or additional data

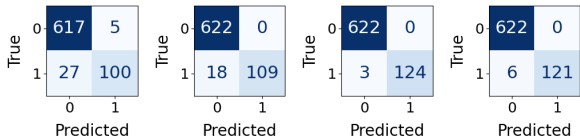

Figure 8: Confusion matrices of the prediction results of GAT on testing data. From left to right: P-O, P-O + coP-P, P-O + simP-P, P-O + coP-P + simP-P.

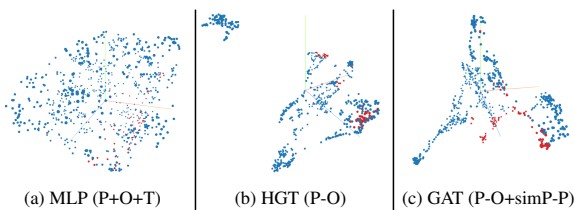

(a) MLP (P+O+T)  (b) HGT (P-O)  (c) GAT (P-O+simP-P)

Figure 9: Visualization of Learned Representation. Red points indicate civil officials with military power, and blue points indicate those without.

sources is apparent to enhance accuracy when data is limited.
- Direct evidence of military power in related narratives significantly improved prediction accuracy, particularly when using P-O + simP-P. This underscores the importance of integrating text-based evidence.
- Abundant co-occurrence relations can sometimes introduce noise, leading to inaccurate predictions. This was particularly evident in cases where officials had numerous related texts, but the co-occurrence did not necessarily indicate the presence of military power. This indicates the need for careful management of such relations in future models.

### 4.5 Embedding Visualization

Using UMAP (McInnes and Healy, 2018) implemented through TensorBoard (Abadi et al., 2015), we projected learned representations into a 3D space (see Figure 9). It is evident that the GAT (P-O+simP-P) model distinguishes well between officials with and without military power (Figure 9c), with the lower-performing HGT (P-O) also showing some differentiation (Figure 9b). Conversely, the MLP (P+O+T) model blends the two categories, indicating its limited discrimination ability (Figure 9a). The visualization affirms the advantage of our graph-based approach for identifying nuanced differences.

## 4.6 Discussion

Overall, the experimental results confirm the effectiveness of our proposed method in detecting civil officials with military power; however, there are two findings we believe worthy of deliberation:

**Chinese Language Model for Initializing Node T** Our decision to utilize the BERT-base-chinese model as the encoder was informed by our prior experience, where in our separate studies of Ming Dynasty texts found that models tailored for classical Chinese do not show significant improvements in tasks related to text classification that relies heavily on the embeddings. We hypothesize this effectiveness might be due to the similarity between the writing style of the Ming Dynasty historical texts and contemporary Chinese. Additionally, since our task is more on similarity and co-occurrence, we think that a large enough pretrained model such as BERT-base-chinese can produce satisfactory results. We envision that a large language model that can properly be trained on both Modern and Classical Chinese would benefit study in this kind.

**More Effective Way to Fuse coP-P and simP-P Interactions** We suspect the decreased performance when using both types of relations comes from the long text spans in coP-P relations. Given that these relations are drawn from entire paragraphs, there might be a weak connection between a person mentioned at the beginning and those mentioned at the end of a long paragraph. We hypothesize that using shorter text spans, like sentences, might work better and reduce noise. However, segmenting the paragraphs into sentences is a challenge in our case because the original records, like most Classical Chinese texts, lack punctuation marks which poses a segmentation challenge, and during our research, we haven't identified a tool that could efficiently address this challenge. We think that future research will look into better ways to combine these P-P interactions, and we'll leave this as a challenge for future study.

## 5 Related Work

**Prosopography** This field of studying common characteristics of a targeted population focuses on the population's shared lives (Stone, 1971; Keats-Rohan, 2007; Gerritsen, 2008): for example, they might hold the same official position, come from a specific status group or economic class, belong to a specific period, or share a connection to a single individual. Following carefully defined selection criteria, prior works (Bol, 2012; Suddaby et al., 2016) model the biographical information in a systematic way to identify a group of individuals for understanding how the members of a target group were interconnected or "operated within and upon the *institutions* — social, political, legal, economic, intellectual — of their time (Keats-Rohan, 2000)."

**Extraction of biographical information from texts and network** Many existing works that also leverage the rich information in biographical databases and historical texts have focused on the extraction of biographical information from texts and network analysis. Liu et al. (2015), Lai (2016), Yan et al. (2020), and Liu et al. (2021) employ the information in biographical databases to annotate texts, and use the annotated texts to train a language model for a named entity recognition task — identifying persons, locations, and official titles. Armand and Henriot (2021) and Zhang et al. (2021) further use these identified entities to construct a latent social network based on co-occurrence of person names. Several works also explore the correspondence network (Weerdt et al., 2016), kinship network (Liu and Wang, 2017), and career trajectory network (Xiong, 2021).

**Graph Neural Network (GNN) and Representation Learning** GNN utilizes graph structures for *message passing*, a process that aggregates a node's local neighborhood information and update the node's representation. Notable GNN architectures have been proposed. Kipf and Welling (2017) propose graph convolutional network (GCN), facilitating efficient message passing and aggregation between neighbors. Hamilton et al. (2018) propose GraphSAGE, an inductive framework using local neighbor sampling and various aggregations operators including mean, LSTM and pooling. Veličković et al. (2018) propose graph attention network (GAT), incorporating attention mechanism into GNNs and allowing different weights for nodes within the same neighborhood. Hu et al. (2020b) propose heterogeneous graph transformer (HGT), employing meta relations in heterogeneous graphs to parameterize shared weight matrices for generalization and integrates with transformer-like attention architecture.

Such GNNs have been utilized in contemporary political figure representation learning. For instance, (Mou et al., 2021) and (Feng et al., 2022)

have applied GNNs to jointly model legislators and learn representations, incorporating voting behavior and public statements or leveraging information from Wikipedia and think tanks, respectively.

Our proposed dataset and method differs from these related work in that we propose to jointly model the officials, the events from both structured and textual data, and leverage representation learning with a graph neural network (GNN) to identify nuanced characteristics of past officials, thus giving a comprehensive view of historical figures not seen before. Additionally, the textual data features a high volume of co-occurrence ties and latent associations, which can be further leveraged to investigate the interpersonal relationships and political subgroups among past officials. Lastly, our approach also unveils text-based evidence that substantiates the identified nuanced characteristics.

# 6    Conclusion and Future Work

In this work, we introduce a dataset, derived from career records of the officials in the Ming Dynasty and their relevant historical texts in *Ming Shi Lu* (明實錄, or *Imperial Annals of Ming Dynasty*). Our *MingOfficial* dataset captures the dynamic social structures among officials throughout the reigns of various Ming emperors. In addition, we took the first step to study the officials in the past with a graph-based representation learning approach.

Our proposed method underscores the potential of integrating the literary context of historical research in a less direct yet effective manner. Instead of depending heavily on the straightforward interpretation of the contextual meaning of historical research, this approach allows for an indirect understanding through relationships that are much easier to extract from the text. This method addresses a significant challenge: minimizing the dependency on labor-intensive interpretation while integrating the rich context from historical texts.

Experiments shows that our approach can learn effective representation to discern nuanced characteristics of individuals such as civil official's military power in the Ming Dynasty, despite being trained on only 49 labeled instances, with just 8 positive cases. This finding highlights the potential to significantly reduce annotation labor in future applications

Despite the advancements achieved, several potential research areas remain unexplored due to the study's focused scope, which we earmark for future work. For example, we envision a future extension on extracting local-scale, fine-grained relationships between officials in targeted groups that are not explicitly documented in historical documents. Another prospective extension is applying our approach to other historical studies, such as examining other governmental systems (for instance, the Byzantine Empire) with complex and multimodal historical recordings. Lastly, we foresee opportunities in leveraging our approach to facilitate various other domains, like sociology and political science, thereby further broadening the applicability and impact of our research.

## Limitations

Despite the demonstrated efficacy of our proposed method in identifying nuanced characteristics within a group of people, there are notable limitations. Firstly, our approach has not been contrasted with other potential methods for identifying target groups. This lack of comparative analysis is primarily due to the scarcity of related work in the field, including no other publicly available datasets that align with *MingOfficial*'s characteristics.

We consider our present effort as an initial demonstration, a proof of concept that paves the way for more extensive research endeavors in the future. While our current approach was tested on a single dataset, it holds potential for wider applications across diverse datasets. To provide more context, our approach is well-suited for probing **historical documents** like Veritable Records of the Joseon Dynasty (朝鮮王朝實錄), Veritable Records of Qing (清實錄), and Chorography (地方誌). Beyond that, we can also delve into **contemporary archives**, especially those that highlight local intricacies and interpersonal dynamics, such as records from Taiwan's Legislature and Hong Kong's Legislative Council. We believe that our current work is just the beginning. It lays the groundwork for wider and more diverse research down the road.

Additionally, a significant drawback of our method is the limited interpretability of our GNN models. The specifics of how the model draws predictions from the data provided in the *MingOfficial* dataset remain unknown. A more transparent understanding of which career records or official relations significantly contribute to the model's predictions would greatly aid researchers in the field. However, such exploration extends beyond the scope of our current study and is a focus for future work.

Furthermore, our study is constrained by the quality and scope of our data. As with many historical databases, our *MingOfficial* dataset might contain inaccuracies and omissions that could affect our results. Moreover, the data is confined to the Ming Dynasty, limiting the generalizability of our findings to other historical periods or cultures. The scalability of our method also remains untested. Although our approach proves effective for the present dataset, its performance with significantly larger datasets or different time frames remains uncertain.

Lastly, our research relies on certain assumptions, including the presumption that all significant characteristics and relationships are captured by the database and textual records. If these assumptions fail, our method might overlook key aspects influencing an official's acquisition of military power. Future work can address these limitations by testing our method on more comprehensive and diverse datasets, as well as exploring mechanisms for model interpretability and validation for assumptions.

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

Ying Zhang, Susan Xue, Zhaohui Xue, and Li Ni. 201. Chinese-English Dictionary of Ming Government Official Titles (明代職官中英辭典).

## A Official Position Categorization in Data Construction

In our data construction process, we organize office (positions) into 123 categories that mirror the administrative structure of the Ming Dynasty government. This categorization groups the titles of core officials and subordinates from different departments into office categories, simplifying the conversion of of career information into a graph structure.

As an illustration, we present the top three most frequently appearing position categories for each emperor in Table 4. Detailed categorization can be found in Figure 14 to 22 at the end of the appendix.

## B Breakdown of the Number of Processed texts by each Emperor in the Ming Dynasty

Table 5 provides a detailed breakdown of the number of processed texts for each emperor during the Ming Dynasty, which spanned from the reign of Hongwu (1368-1398) to Chongzhen (1627-1644). For each emperor, the table specifies:

- The name of the emperor (both in English and traditional Chinese characters)

- The reign period of each emperor in A.D.

- The total number of texts processed during their reign

- The total number of official names identified from the texts of their respective reigns

## C Details on Exploratory Analysis of *MingOfficial*

**Insight 1: *MingOfficial* Reflects the Changing Power Dynamics among Major Political Groups Over Time.** Our initial analysis illustrates the dataset's ability to trace the historical shifts in power among major political groups. We aim to observe the changing prominence of civil officials, military officers, and eunuchs throughout the Ming Dynasty.

We employed a term frequency-document frequency (tf-df) calculation, inspired by the numerical statistic term frequency-inverse document frequency (tf-idf) (Sparck Jones, 1988), where a term corresponds to an office,

$$\text{tfdf}(t, d, D_y) = \text{tf}(t, d) * \text{df}(t, D_y) \quad (5)$$

where the term frequency $\text{tf}(t, d)$ is the frequency of term within a record, $\text{df}(t, D_y)$ is a measure of how much influence an office provides within a year

$$\text{tf}(t, d) = f_{t,d} \quad (6)$$

$$\text{df}(t, D_y) = \log\left(\frac{|d \in D_y : t \in d|}{N}\right) \quad (7)$$

with $t$ = office mention, $d$ = record, $D_y$ = set of records with in year $y$, and $N = |D_y|$. In practice, $df(t, D)$ is computed as:

$$\log\left(\frac{|d \in D_y : t \in d|}{N + 1}\right) + 1 \quad (8)$$

Instead of incorporating the inverse document frequency to diminish the frequency of terms that not only frequently occur in a record but also frequently in all of the records within a year, we compute the document frequency to strengthen it. We reason that the more occurrence of an office in multiple records, the more important an office is in the year. To observe the change in importance of different offices over time, we rank each office in a record by tf-df and count the total number of times each office ranks first by year.

The result, displayed in Figure 6, reveals a significant trend: in the early Ming, military officers had significant influence. However, by the late 15th century, the status of civil officials gradually improved and eventually surpassed the others. Moreover, eunuchs also held considerable influence during certain periods.

Overall, this graph showcases the dynamic power shifts between the three major political groups over time. In particular, it provides a preliminary explanation of the phenomena whereby military officers were heavily relied upon in the early Ming dynasty, and how the power in military discourse gradually shifted towards civil officials and eunuchs during the middle period. This insight suggests that the *MingOfficial* dataset is a valuable resource for understanding the evolving power dynamics in the Ming dynasty. Detailed analysis of these trends is left as future work.

**Insight 2: *MingOfficial* Preserves the Intricate Social Relations among Officials.** To further demonstrate the dataset's capacity for depicting complex social relations, we construct co-occurrence networks based on the person mentions

| Emperor | No.1 | No.2 | No.3 |
|---|---|---|---|
| Hongwu | Noble Titles | Core Officers in the Guard Military Commands | Other Officers in the Chief Military Commissions |
| Jianwen | Core Officers in the Regional Military Commission | Other Officers in the Chief Military Commissions | Noble Titles |
| Tainshun | Core Officers Provincial in the Administration Commission | Hanlin Academy | Noble Titles |
| Chenghua | Core Officers Provincial in the Administration Commission | Other Officers in the Censorate | Sub-prefectural Officers |
| Hongzhi | Core Officers Provincial in the Administration Commission | Hanlin Academy | Sub-prefectural Officers |
| Zhengde | Core Officers Provincial in the Administration Commission | Other Officers in the Censorate | Provincial Surveillance Commission |
| Jiajing | Core Officers Provincial in the Administration Commission | Other Officers in the Censorate | Provincial Surveillance Commission |
| Taichang | Other Officers in the Censorate | Grand Secretary | Core Officers Provincial in the Administration Commission |
| Tianqi | Other Officers in the Censorate | Three Preceptors and Three Juniors in the Eastern Palace | Core Officers Provincial in the Administration Commission |
| Chongzhen | Other Officers in the Censorate | Sub-prefectural Officers | Grand Coordinator |

Table 4: The top three most frequently appearing position categories under each specific emperor in *MingOfficial*.

| Book of Emperor | Reign Period (A.D.) | # Paragraphs | # Officials |
|---|---|---|---|
| Hongwu (洪武帝) | 1368-1398 | 2,687 | 451 |
| Yongle (永樂帝) | 1402-1424 | 2,032 | 547 |
| Hongxi (洪熙帝) | 1424-1425 | 227 | 199 |
| Xuande (宣德帝) | 1425-1435 | 2,447 | 496 |
| Zhengtong (正統帝)[1] | 1435-1464 | 5,715 | 970 |
| Chenghua (成化帝) | 1464-1487 | 5,742 | 1,024 |
| Hongzhi (弘治帝) | 1487-1505 | 4,142 | 1,000 |
| Zhengde (正德帝) | 1505-1521 | 5,042 | 1,258 |
| Jiajing (嘉靖帝) | 1521-1567 | 11,197 | 2,042 |
| Longqing (隆慶帝) | 1567-1572 | 2,182 | 735 |
| Wanli (萬曆帝) | 1572-1620 | 15,689 | 1,721 |
| Taichang (泰昌帝) | 1586-1620 | 217 | 220 |
| Tianqi (天啓帝) | 1620-1627 | 5,229 | 1,012 |
| Chongzhen (崇禎帝)[2] | 1627-1644 | 4,494 | 1,230 |

Table 5: Processed Text Count per Emperor: A tally of texts processed for each emperor during the Ming Dynasty.
[1]Including the reign of Jingtai (景泰帝, 1449-1457) and Zhengtong's second term under regnal name Tianshun (天順, 1457-1464). [2]Unlike books of all other emperors, this book was compiled after the fall of Ming Dynasty.

in texts and visualize the network with Gephi (Bastian et al., 2009), a network analysis tool. Figure 7 present these networks, which span across the reigns of two different emperors from 1449 to 1451. Yu Qian is prominently featured in both

(highlighted with a red circle).

The distinct communities within the network are indicated by node colors, as detected by the Louvain method built into Gephi. Text colors within nodes represent different personnel types: blue for

civil officials, purple for military officers, green for eunuchs, and orange for the imperial family. The depicted variations in network structures between the two periods offer unique insights into the dynamics of interpersonal relationships and social structures over time.

In summary, the *MingOfficial* dataset serves as a potent resource for exploring not only individual biography and prosopography, but also broader social dynamics such as class, mobility, and political networks. Our examination of potential associations and network changes underscore the intricate social interactions preserved within the dataset, emphasizing its value for historical and sociological studies.

## D Historical Background

In this section, we will give a brief overview of the governmental system in the Ming dynasty and the interaction of officials operating within it.

### D.1 The Ming Government System

The governmental system matured in the Ming times is the culmination of the development of the bureaucratic system in the previous dynasties, generally regarded as the epitome of the bureaucratic system in Chinese history (Hucker, 1958). The organization of the system has three tiers in general. The top tier comprises 10 clusters of government offices in the central government, the second has 48 clusters of government agencies, and the third has 230 departments (Zhang et al., 201). Besides abolishing a few top-tier political institutions in the central government to concentrate power in his own hands, the founding emperor of the Ming dynasty (1368-1398), Zhu Yuanzhang, also established new government organs according to the social and political needs of the time, forming an organizational structure "largely perpetuated with admiration by the succeeding Ching dynasty (Hucker, 1998)."

### D.2 The Interplay of the Political Groups in the Ming Government

Three major political groups operated within the Ming government — civil officials, military officers, and eunuchs. The Ming emperors adopted a few measures to prevent one group from superimposing the others.

To limit the power of the military officers and institutions, the Ming emperors separated the military leaders from their soldiers by keeping the military leadership — the authority to manage and command the troops — under the control of a small number of high-ranking civil officials. (Filipiak, 2012) As the civil officials are thoroughly indoctrinated in the ritual and principles of good government written in the Confucian literature, they were less likely to challenge the imperial power, thus strengthening the military governance by the emperor. (Zhang, 2011; Robinson, 2017) Moreover, to alleviate the tension between the Ming emperors, and the civil officials and military officers, eunuchs were also assigned to various influential tactical military positions. (Li, 2018) As time progressed, these measures created tension between the three groups, contributing to the country's ills and hastening the decline of the Ming dynasty.

## E Resource-intensive Annotation for Determining Civil Officials' Military Power

Determining whether a civil official had actual military power during their career is a complex task that requires a significant investment of time and resources. The process cannot solely rely on career records in the database of names and biographies (人名權威人物傳記資料庫)(IHP), as these records often lack sufficient detail. For instance, while many Ministers of War led armies in battle, this was not a uniform practice across all reigns during the Ming Dynasty.

Given these challenges, the process for making reliable annotations involves a detailed examination of an official's career trajectory, changes in their roles, and any evidence of their exercising military authority. This extensive process includes:

- Conducting in-depth reviews and studies of a wide range of historical literature.

- Verifying evidence by cross-referencing various historical sources.

- Performing rigorous historical analyses.

This procedure requires both expertise in the subject matter and a considerable investment of time. For this study, the development of the annotation guideline and the annotation of civil officials with military power took us approximately three months.

## F Implementation Details for Official Representation Learning

### F.1 Node Initialization

We encode the texts with the an open-source weight of Chinese BERT model (cki) and Hugging-face (Wolf et al., 2020) framework , yielding an initial embedding dimension of 768 for text nodes. We then implemented the metapath2vec model with PyTorch-Geometric (Fey and Lenssen, 2019) to produce a 64 dimension representation with following settings: walk length at 40, context size at 8, walks per node at 20, number of negative samples at 1, The model was trained for 20 epochs with a batch size of 32.

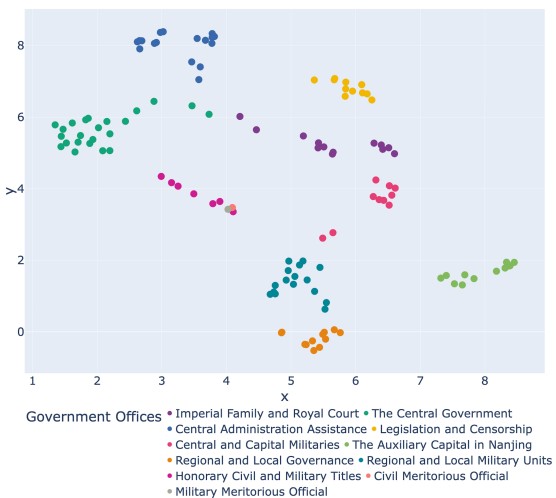

Government Offices
- Imperial Family and Royal Court • The Central Government
- Central Administration Assistance • Legislation and Censorship
- Central and Capital Militaries • The Auxiliary Capital in Nanjing
- Regional and Local Governance • Regional and Local Military Units
- Honorary Civil and Military Titles • Civil Meritorious Official
- Military Meritorious Official

Figure 10: The official position embeddings learned by the metapath2vec model with a clear clustering aligned with the actual office classification.

The initial official position embeddings, visualized in Figure 10, illustrates that official positions within the same offices (points of identical colors) cluster together, signifying that the official position embeddings effectively represents the administrative structure of the Ming Dynasty.

### F.2 Model Implementation and Hyperparameters

We implemented GNN and Multi-layer Perceptron (MLP) models using PyTorch (Paszke et al., 2019) and PyTorch-Geometric. All models use Adam optimizer (Kingma and Ba, 2014), with $\beta_1 = 0.9$, $\beta_2 = 0.999$, $\epsilon = 1e - 08$, and a batch size 8.

The MLP model, with a learning rate set to $1e - 3$, took as input a concatenation of the official's own representation and the averaged representations of the position (office) and/or text re-

lated to the official. It consisted of a normalization layer followed by two projection layers, predicting whether the civil official held military power.

For GNN models, we use a learning rate of $1e-4$ for stage 1, and $1e - 3$ for stage 2. We set $k = 20$ for constructing the simP-P relation. The models consisted of 2 layers with a hidden-layer dimension of 128. Specifically, for HGT and GAT models, we set the number of attention heads to 4. Dropout and early stopping strategies were employed to prevent overfitting. The specific number of training epochs is given in Table 3.

### F.3 Handling Class Imbalance

As shown in Table 2, our dataset suffered from extreme class imbalance. To address this, we utilized an imbalanced sampler implemented in PyTorch-Geometric (Fey and Lenssen, 2019) in both stages of representation learning (see 3.2). In stage 1, we upsampled the eunuch and military officer sets to match the number of civil officials. In stage 2, we upsampled the set of civil officials with military power to equal the number of civil officials without military power. This sampling process was applied to both our training and validation sets.

## G Error Analysis

We conducted an error analysis on the best performing GNN model – GAT, and extracted several important findings. Confusion matrices for the prediction results under different settings are displayed in Figure 8.

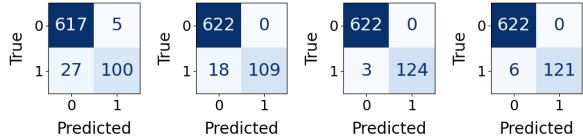

Figure 11: Confusion matrices of the prediction results of GAT on testing data. From left to right: P-O, P-O + coP-P, P-O + simP-P, P-O + coP-P + simP-P.

**Identification of Nuanced Characteristics, like Military Power Among Civil Officials, is Insufficient with Merely Career Records.** The GAT model, despite its strong performance across all settings in predicting Civil Officials without Military Power (CO w/o MP), incorrectly predicts five false positives when relying only on P-O. These false positives, as exemplified by Zhang Ben (張

| No. | Career records | Office Position Category | Reign Period |
|---|---|---|---|
| 1 | Right Vice Censor-in-chief in the Censorate (都察院右副都御史) | Other Officers in the Censorate (都察院其他職員) | Hongzhi(弘治)-Zhengde(正德) |
| 2 | Secretary of the Bureau of Construction in the Ministry of Works at Nanjing (南京工部營繕司主事) | Other Officers in the Ministry in Nanjing (南京六部門其他職員) | Chenghua (成化) |
| 3 | Vice Director of the Bureau of Construction in the Ministry of Works at Nanjing (南京工部營繕司員外郎) | Other Officers in the Ministry in Nanjing (南京六部門其他職員) | Chenghua (成化) |
| 4 | Prefect at Xiangyang (襄陽府知府) | Prefectural Officers (府官) | Chenghua (成化) |
| 5 | Prefect at Qingzhou (青州府知府) | Prefectural Officers (府官) | Chenghua (成化) |
| 6 | Right Administration Vice Commissioner of a Province at Guangdong (廣東右參政) | Core Officers Provincial in the Administration Commission (承宣布政主要職員) | Hongzhi (弘治) |
| 7 | Right Provincial Administration Commissioner in the Provincial Administration Commissionat Huguang (湖廣布政司右布政使) | Core Officers Provincial in the Administration Commission (承宣布政主要職員) | Hongzhi (弘治) |
| 8 | Grand Coordinator at Jiangxi (江西巡撫) | Grand Coordinator (巡撫) | Hongzhi (弘治)-Zhengde (正德) |

Table 6: Zhang Ben's Career Records: A list of positions held by Zhang Ben(張本), with positions often associated with military power, such as Officers Provincial in the Administration Commission (No.6 and 7) and Grand Coordinator (No. 8).

本) in Table 6, depict career trajectories that resemble those of CO w/ MP. They hold positions often associated with military power, such as Grand Coordinator (巡撫), Supreme Commander (總督), Military Superintendent (提督), Minister of War (兵部尚書), Vice Commander-in-chief (協理京營戎政), Officers Provincial in the Administration Commission (承宣布政相關職員), and Officers in the Censorate (都察院相關職員). We conjecture that this similarity leads model to erroneous predictions.

However, the model's performance improved considerably when it incorporated additional person-to-person information, such as co-occurring positions (coP-P), similarity of individual embeddings (simP-P), or both. Consequently, false positives for CO w/o MP were effectively eradicated. This outcome highlights the inadequacy of relying solely on career records to discern nuanced characteristics, such as military power among civil officials.

Notably, true positive predictions using P-O were associated with an average of 23.7 career records, compared to a mere average of 7.5 records for false negatives. This discrepancy underscores the potential value of comprehensive career records in enhancing model accuracy, particularly when the data is confined to career records alone.

**Sparse Information on Individuals Inhibits Accurate Predictions.** Across all four settings, the model consistently misclassified three specific individuals as false negatives. These individuals had limited data, with no more than two career records and between one and five related texts. Furthermore, none of these texts presented direct evidence of the individuals possessing Military Power (MP). This finding emphasizes the need for abundant individual-level information to ensure accurate predictions. Therefore, enriching the dataset or leveraging additional data sources might enhance prediction accuracy when the available data is limited.

**Direct Evidence of Military Power in Related Narratives Enhances SimP-P Efficacy.** Examining the true positive predictions made using official positions and individual embedding similarity (P-O + simP-P) revealed that related texts often supplied direct evidence of the officials having military power, especially when other settings (P-O, P-O + coP-P) fell short. An illustrative example includes a narrative stating, "Supervising Secretary Dai Bain (戴弁) and six others who were decreed to be stationed at the border. (給事中戴弁等七人各賜曰爾等分邊).", confirming Dai Bain (戴弁)'s military power. This evidence substantiates the effectiveness of simP-P in deriving meaningful associations from textual content, further highlighting the added value of integrating text-based evidence into predictive tasks.

| No. | Career records | Office Position Category | Reign Period |
|---|---|---|---|
| 1 | Supervising Secretary in Scrutiny for Revenue (戶科給事中) | Office of Scrutiny for Revenue (戶科) | Jiajing (嘉靖) |
| 2 | Secretary in the Ministry of Justice (刑部主事) | Other Officers in The Ministry or Justice (刑部其他職員) | Jiajing (嘉靖) |
| 3 | Vice Minister in the Seals Office (尚寶司少卿) | Other Officers in Secretary Offices (秘書門其他職員) | Jiajing (嘉靖) |
| 4 | Vice Minister in the Seals Office (尚寶司卿) | Other Officers in Secretary Offices (秘書門其他職員) | Jiajing (嘉靖) |
| 5 | Assisant Minister in the Court of Imperial Entertainments(光祿寺寺丞) | Court of Imperial Entertainments (光祿寺) | Jiajing (嘉靖) |
| 6 | Vice Governor in the Yingtian Superior Prefecture (應天府府丞) | Other Officers in the Yingtian Superior Prefecture (應天府其他職員) | Jiajing (嘉靖) |

Table 7: Sun Yunzhong's Career Records: A record of positions held by Sun Yunzhong (孫允中), demonstrating a lack of roles typically associated with military power among civil positions.

**Abundant Co-occurrence Relations Could Introduce Noise.** A clear manifestation of this issue arises in three instances of false negative predictions observed under the settings of P-O + coP-P + simP-P, P-O + coP-P, and P-O alone. Two of these erroneous predictions involve Liu Jingshao (劉景韶) and Song Ying-Chang (宋應昌), each with fewer than three career records. The third case, Sun Yunzhong(孫允中), possesses six career records, but significantly deviates from a typical CO w/ MP, as demonstrated in Table 7. The model's incorrect predictions under the P-O alone setting can be attributed to these atypical career records.

However, the P-O + coP-P + simP-P configuration presents a different problem, as the model still fails to predict correctly despite the availability of related texts that provide direct evidence of MP. This issue is exemplified by Song Ying Chang's co-occurrence network (Figure 13). Despite military-related narratives present in Song Ying Chang's textual data (Figure 12), an overload of co-occurrence relations introduced by coP-P causes the model to overlook his military power. This underscores the need for future models to employ more refined management or filtering of co-occurrence relations, in order to minimize noise and improve predictive accuracy.

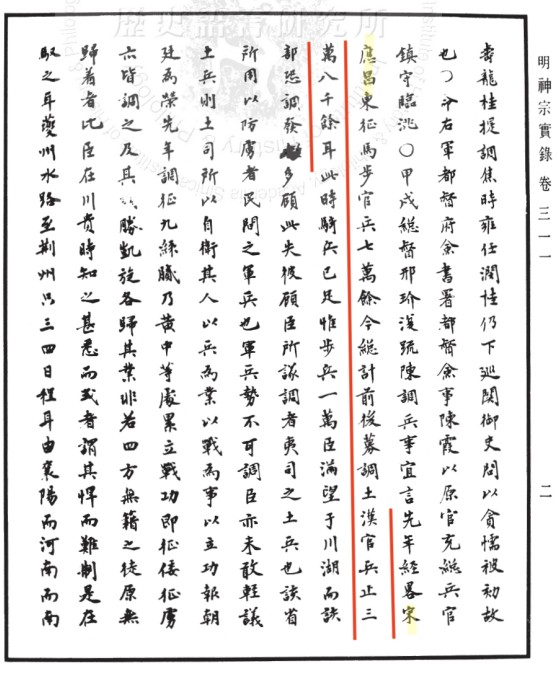

Figure 12: Textual Evidence of Military Power: An excerpt from the text providing evidence of military power for Song Ying-Chang (宋應昌). The underlined text in red translates to: 'In previous years, Pacification Commissioner Song Ying-Chang (宋應昌) was sent on an eastern campaign with more than 70,000 cavalry and infantry soldiers. However, more recently, the total number of locally recruited Chinese official soldiers only amounts to over 38,000."

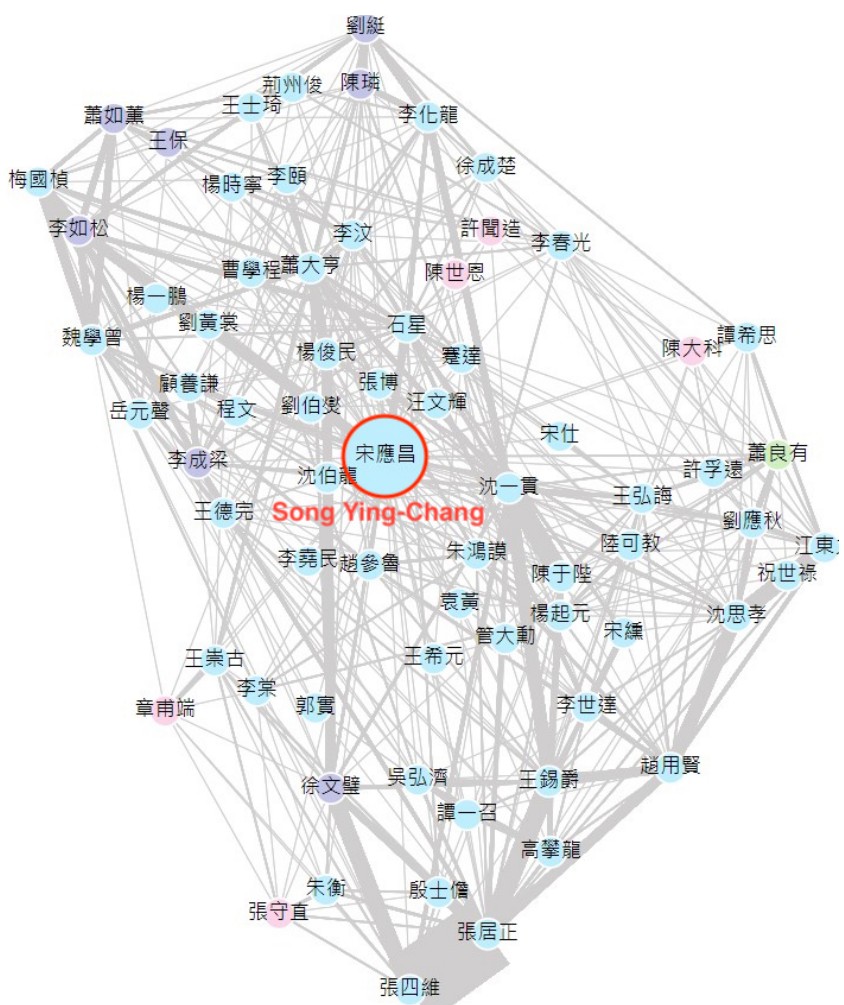

Figure 13: Co-occurrence Network of Song Ying-Chang: A graphical representation of Song Ying-Chang's (宋應昌) network, as depicted in 68 relevant historical texts from *Ming Shi Lu*. Civil officials, military officers, and non-categorizable individuals are represented as blue, purple, and pink nodes, respectively. The edge thickness indicates the frequency of co-mentions in historical texts.

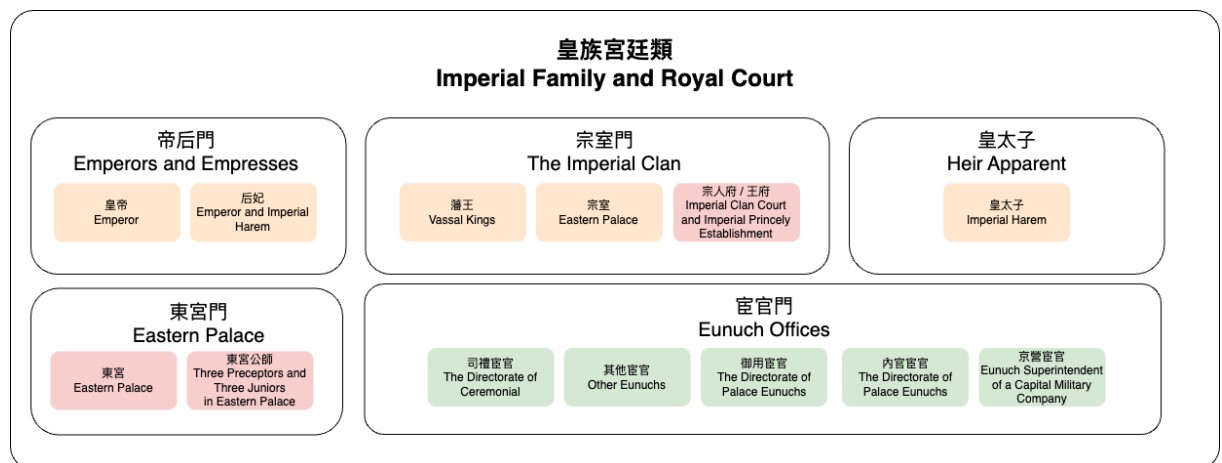

Figure 14

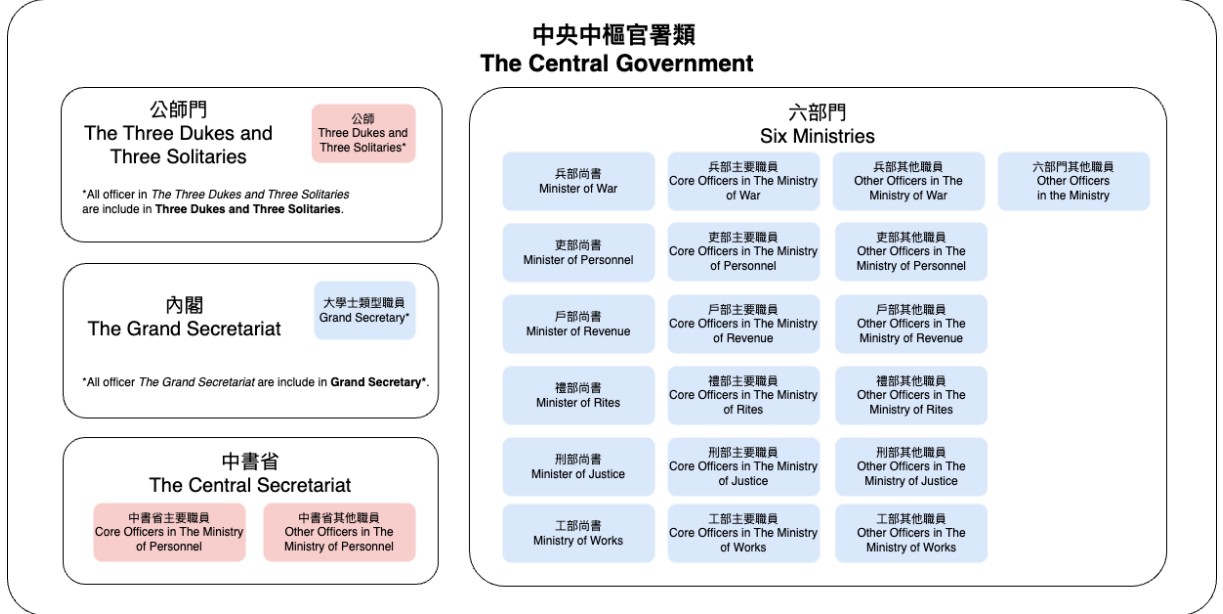

Figure 15

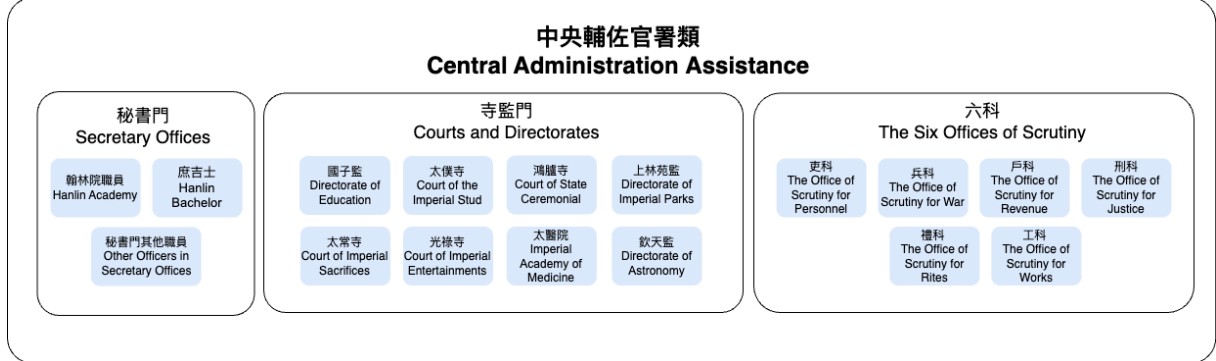

Figure 16

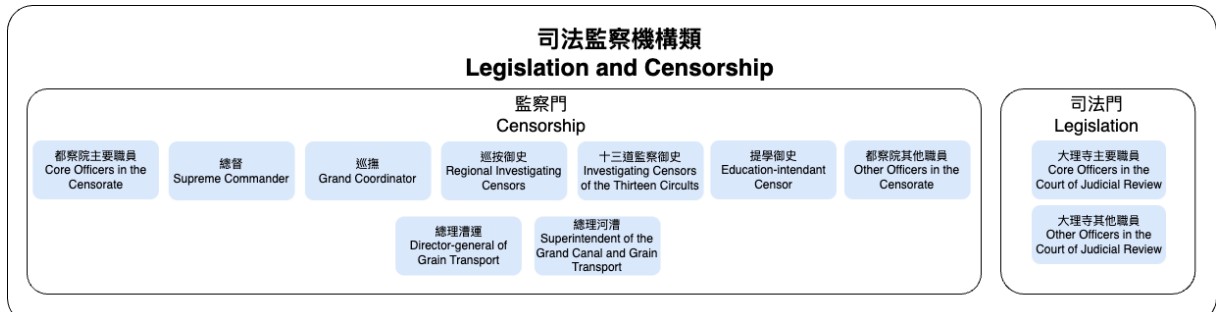

Figure 17

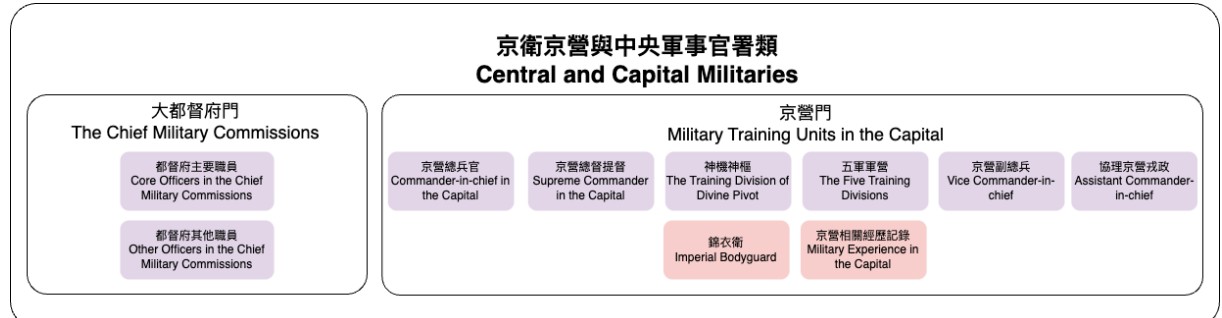

Figure 18

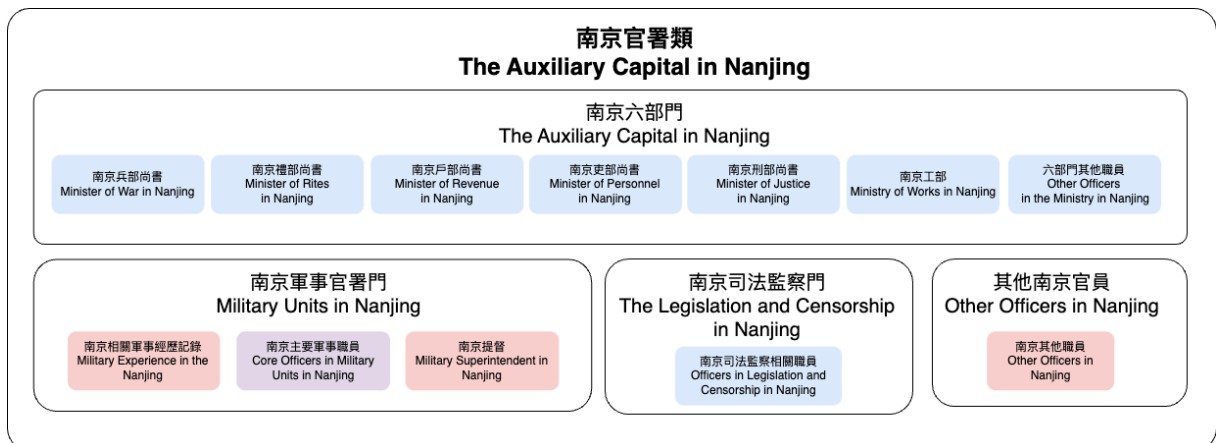

Figure 19

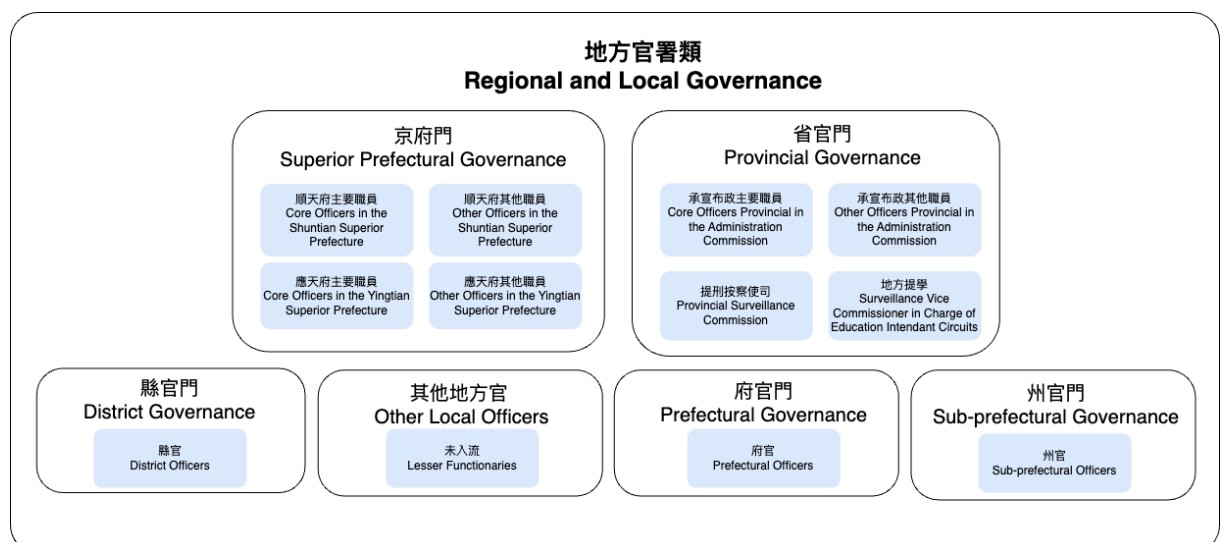

Figure 20

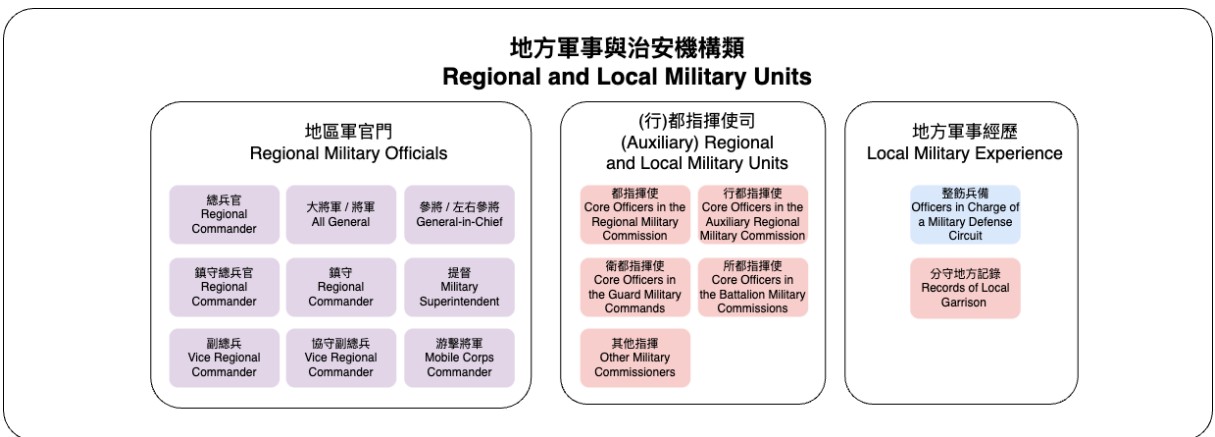

Figure 21

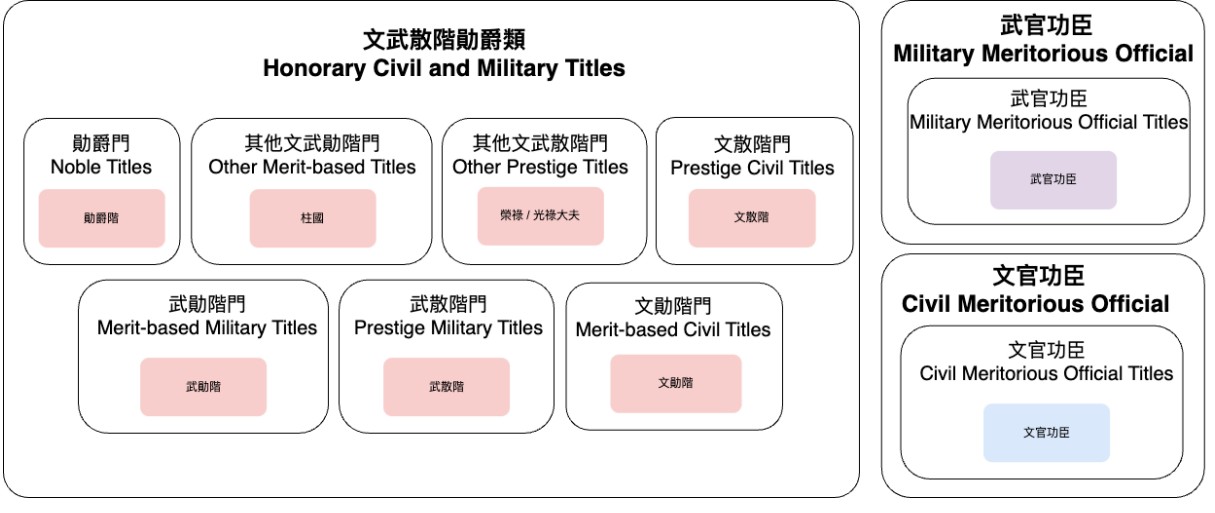

Figure 22