# OpenReview forum: "MingOfficial: A Ming Official Career Dataset and a Historical Context-Aware Representation Learning Framework"
_EMNLP/2023/Conference — EMNLP 2023 Main_

### Official Review · Reviewer_zST9 · 2023-08-01

**Soundness:** 4

**Excitement:**

3: Ambivalent: It has merits (e.g., it reports state-of-the-art results, the idea is nice), but there are key weaknesses (e.g., it describes incremental work), and it can significantly benefit from another round of revision. However, I won't object to accepting it if my co-reviewers champion it.

**Paper Topic And Main Contributions:**

The paper introduces MingOfficial, a multi-modal dataset focusing on officials from China's Ming Dynasty. The authors further propose a GNN-based method to investigate social structures and provides powerful features for downstream tasks. The experiments verify the proposed method can identifying nuanced identities.

**Reasons To Accept:**

1. The paper addresses a fascinating and essential topic in the field of Chinese studies. This topic is of significant interest to scholars and researchers in history and linguistics.
2. The paper provides sufficient information to reproduce the experimental results.

**Reasons To Reject:**

1. The paper's reliance on only one dataset could be perceived as a potential limitation, as it may hinder the ability to establish the universality or general applicability of the proposed method.
2. The proposed method appears to be primarily built upon existing research, and it lacks sufficient innovation.

**Reproducibility:**

5: Could easily reproduce the results.

**Reviewer Confidence:**

4: Quite sure. I tried to check the important points carefully. It's unlikely, though conceivable, that I missed something that should affect my ratings.

---

> ### Author Rebuttal · Authors · 2023-08-28
>
> We thank the reviewer for their time, substantial feedback and insightful questions. We also thank the reviewer for recognizing our paper "addresses a fascinating and essential topic in the field" and  provides sufficient information to reproduce."
>
> We would like to address the reviewer’s concerns as follows, which will also be added to the revised manuscript:
>
> # Regarding the innovation and the big picture:
>
> We agree that work indeed leverages largely a wide range of existing techniques, yet we argue that its novelty lies in the unified application of advanced technical tools to a domain not traditionally addressed with such methodologies. We aim to introduce a new paradigm for addressing similar questions in digital humanities and historical research as follows.
>
> Concretely, our proposed method underscores the potential of integrating the literary context of historical research in a less direct yet effective manner. Instead of depending heavily on the straightforward interpretation of the contextual meaning of historical research, this approach allows for an indirect understanding through relationships that are much easier to extract from the text. This method addresses a significant challenge: integrating the rich context from historical texts while minimizing the dependency on labor-intensive interpretation. By leveraging elements that can effectively represent the features of the context, this approach provides a new, efficient avenue for uncovering profound insights into our past.
>
> Furthermore we envision that this new paradigm will fully materialize its potentials when applied to a wide range of historical and modern records/tasks, as we detailed in the paragraph below.
>
> # Regarding the limitation on one dataset and the future works.
>
> We thank both reviewer ZgKy and zST9 for pointing out this concern, and we put our response to the valuable concern here.
>
> We sincerely acknowledge the concerns raised about limiting our experiments to the MingOfficial dataset. Our choice was primarily influenced by the unique nature of the MingOfficial dataset. Through our research, we found no other publicly available datasets that align with MingOfficial's characteristics, which underscores its singular presence in the domain.
>
> Our primary goal is to introduce a new paradigm for addressing similar questions in digital humanities and historical research, emphasizing the unified application of state-of-the-art technical tools in areas previously overlooked. We consider our present effort as an initial demonstration, a proof of concept that paves the way for more extensive research endeavors in the future.
>
> While our current approach was tested on a single dataset, it holds potential for wider applications across diverse datasets. To provide more context, our approach is well-suited for probing **historical documents** like Veritable Records of the Joseon Dynasty (朝鮮王朝實錄), Veritable Records of Qing (清實錄), and Chorography (地方誌). Beyond that, we can also delve into **contemporary archives**, especially those that highlight local intricacies and interpersonal dynamics, such as records from Taiwan's Legislature and Hong Kong's Legislative Council. We believe that our current work is just the beginning. It lays the groundwork for wider and more diverse research down the road.
>
>
> # Regarding the typos and presentational issues:
>
> We sincerely thank the reviewer for raising these issues. We will revise the manuscript to correct presentational issues, including typos, grammar errors, missing references to Figures and the incomplete line at the end of the limitation section.

---

### Official Review · Reviewer_qxZu · 2023-08-05

**Soundness:** 4

**Excitement:**

4: Strong: This paper deepens the understanding of some phenomenon or lowers the barriers to an existing research direction.

**Paper Topic And Main Contributions:**

This paper proposes MingOfficial, a dataset of official careers in the Ming Dynasty for historical analysis. The dataset contains both structured data and unstructured text for representation learning. Experiments demonstrate that the learned representations with the MingOfficial dataset do demonstrate certain properties of historical significance.

**Reasons To Accept:**

+ representation learning of historical people is an interesting research question
+ the MingOfficial dataset could be a useful resource

**Reasons To Reject:**

- Is the base language model used to encode historical text actually trained on ancient mandarin? How did the authors evaluate this adopted language model?

- Why is military power selected as a special attribute for representation learning? It would be nice to have an ablation study removing this learning objective or adding other attributes as part of the learning signal.

- There is no Figure 11 in the paper, but line 434 indicates otherwise. In addition, there is a ?? on line 436.

- I suggest having an ethics statement to discuss relevant concerns if any. Also, the limitations section is incomplete on line 629.

**Reproducibility:**

4: Could mostly reproduce the results, but there may be some variation because of sample variance or minor variations in their interpretation of the protocol or method.

**Reviewer Confidence:**

4: Quite sure. I tried to check the important points carefully. It's unlikely, though conceivable, that I missed something that should affect my ratings.

---

> ### Author Rebuttal · Authors · 2023-08-28
>
> We thank the reviewer for their time, substantial feedback and insightful questions. We also thank the reviewer for recognizing our paper is on "an interesting research question" and its dataset "could be a useful resource".
>
> We would like to address the reviewer’s concerns as follows, which will also be added to the revised manuscript:
>
> # Regarding the Chinese language model used
>
> The base language model we employed for encoding historical text was not specifically trained on Classical Chinese. At the time of our research, we were unable to identify a well-established model dedicated to Classical Chinese. Our decision to utilize the BERT-base-chinese model as the encoder was informed by our prior experience, where in our separate studies of Ming Dynasty texts found that models tailored for classical Chinese do not show significant improvements in tasks related to text classification that relies heavily on the  embeddings.
>
> We hypothesize this effectiveness might be due to the similarity between the writing style of the Ming Dynasty historical texts and contemporary Chinese. Additionally, since our task is more on similarity and co-occurrence, we think that a large enough pretrained model such as BERT-base-chinese can produce satisfactory results.
>
> Yet, we fully agree that this issue deserves further discussion. We envision that a large language model that can properly be trained / fine-tuned on both Modern and Classical Chinese would benefit study in this kind.
>
>
> # Regarding choice of military power as attributes:
>
> The motivation behind our choice of military power among civil officials as a special attribute in representation learning is two-fold --- as it's both an important domain-specific problem and a technically valuable challenge.
>
> First and foremost, the phenomenon of military power among civil officials is an example of well-studied subjects in historical research yet a challenging one. Accurately identifying attributes like military power among civil officials is both time-consuming and demands specialized domain knowledge, often requiring extensive reading of relevant documentation about the person in question. By focusing on this attribute as an example, we believe our method holds potential in uncovering previously unidentified individuals with this attribute, thereby enriching historical research.
>
> Futhoremore, identifying military power among civil officials is a valuable task from a technical point of view and is a showcase of an efficient method capable of detecting attributes not immediately evident from the career records. Concretely, this task requires that a figure representation should (1) be able to mirror the multifaceted nature of these figures, capturing subtler aspects of their roles and power dynamics, (2) properly handle example with contrastive properties, since holding military power is in stark contrast with civil service associated administrative behaviors. In a nutshell this task presents the models with a rigorous test to showcase the model's learning capability.
>
> # Regarding future work:
>
> As we also detailed in the response to Reviewer ZgKy and zST9, although the primary focus of this study centers on a specific dataset, we see our present study as a pioneering step, setting a new paradigm that could be applied to other scenarios which can be future works. The methodology we've developed is well-suited for probing probing **historical documents** like Veritable Records of the Joseon Dynasty (朝鮮王朝實錄), Veritable Records of Qing (清實錄), and Chorography (地方誌) and  **contemporary archives**, especially those that highlight local intricacies and interpersonal dynamics, such as records from Taiwan's Legislature and Hong Kong's Legislative Council. We believe that our work could pave the way for wider and more diverse research down the road.
>
>
> # Regarding the suggestion for an ablation study:
>
> We agree with the reviewer that exploring the impact of removing or integrating other attributes in the learning objective would be insightful. We earmarked such analyses for upcoming studies  due to space constraints in this paper.
>
> # Regarding the typos and presentational issues:
>
> We sincerely thank the reviewer for raising these issues. We will revise the manuscript to correct presentational issues, including typos, grammar errors, missing references to Figures and the incomplete line at the end of the limitation section.

---

### Official Review · Reviewer_ZgKy · 2023-08-05

**Typos Grammar Style And Presentation Improvements:** 1.	Line 154
**Soundness:** 3

**Excitement:**

4: Strong: This paper deepens the understanding of some phenomenon or lowers the barriers to an existing research direction.

**Paper Topic And Main Contributions:**

This paper introduced a new dataset and GNN-based representation learning framework to study the problem of detecting China’s Ming civil officials with military power (CO w/ MP). For the dataset, it consists of career records and personnel types annotations of Ming officials. And in the proposed framework, the authors first constructed a graph with multiple relations among officials, positions and relevant historical texts, and then applied GNN models to encode this information to predict whether an official has military power. Data analysis and experiments are conducted to introduce the dataset and evaluate the proposed framework.

**Questions For The Authors:**

1.	Could the authors better explain the motivation of stage1 training? Moreover, why is only P-O relation used in stage1 but not include other relations? Have the authors ever tried different ways to include the personnel type categorization task, such as joint training with the detection task?

**Reasons To Accept:**

1.	This paper will publish a new dataset containing Ming dynasty officials’ career records, annotated personnel types, and related historical texts, which can benefit the following research work in this field.

2.	This work introduced how to process and annotate the proposed dataset, which offered a guideline to make annotations for other similar tasks. Also, the exploratory analysis can help understand this dataset better.

3.	The authors provided a graph construction method to present Ming dynasty’s complex political landscape, which can be also referred to in other data-driven Chinese historical research studies.

4.	The authors conducted extensive experiments to evaluate the proposed GNN-based framework.


**Reasons To Reject:**

1.	This manuscript needs to be carefully revised in several aspects, including completing the discussion of limitations and addressing various spelling errors (see Typos Grammar Style below).

2.	From the experiment results, it can be found that incorporating coP-P and simP-P always negatively impacted the performance. So, I think there is a need to find a more effective way to fuse these two views of P-P interactions.

3.	The experiments were conducted on only one dataset. It would be better if the evaluation of the proposed framework could be performed on more datasets. The scale of the introduced dataset is relatively small, with a more limited number of annotated data available. However, I understand the difficulty to build such an interesting dataset.


**Reproducibility:**

4: Could mostly reproduce the results, but there may be some variation because of sample variance or minor variations in their interpretation of the protocol or method.

**Reviewer Confidence:**

3: Pretty sure, but there's a chance I missed something. Although I have a good feel for this area in general, I did not carefully check the paper's details, e.g., the math, experimental design, or novelty.

---

> ### Author Rebuttal · Authors · 2023-08-28
>
> We thank the reviewer for their time, substantial feedback and insightful questions. We also thank the reviewer for recognizing that our paper “can benefit the following research work in this field,” "offered a guideline to make annotations for other similar tasks" and "can be also referred to in other data-driven Chinese historical research studies."
>
> We would like to address the reviewer’s concerns as follows, which will also be added to the revised manuscript:
>
> # Regarding the motivation of and the relation used in stage 1 training:
>
> The primary motivation behind stage 1 training is to equip the model with a robust understanding of the distinct roles and functionalities of the three major political groups during the Ming dynasty: military officers, civil officials, and eunuchs. The essence of these functionalities is predominantly determined by the official positions held by individuals within these groups. Therefore, the P-O relation, which encapsulates all the positions an individual has held, is a sufficient feature for this stage of training.
>
> During our preliminary experiments, we also explored using both P-O and P-P relations in stage 1. However, our findings revealed that integrating the P-P relation in stage 1 led to a slower learning curve for categorization without gain. We also believe that introducing external relations in stage 1 might lead to potential misclassifications --- An illustrative scenario might be a eunuch involved in military tasks being mistakenly categorized as a military officer due to the nature of his activities. Consequently, we chose to solely focus on the P-O relation in stage 1, and in doing so, we set a solid foundation, free from ambiguity, upon which subsequent stages can build.
>
> We did not try jointly training the personnel categorization task and the detection task, but we believe this could be left as a insightful future follow-up .
>
> # Regarding coP-P and simP-P and their interaction on performance
>
> We acknowledge and agree with the reviewer's observation regarding the incorporation of coP-P and simP-P and the subsequent negative impact on performance. We also believe that a better method to fuse these two views of P-P interactions would be beneficial.
>
> We suspect the decreased performance when using both types of relations comes from the long text spans in coP-P relations. Given that these relations are drawn from entire paragraphs, there might be a weak connection between a person mentioned at the beginning and those mentioned at the end of a long paragraph.
>
> We hypothesize  that using shorter text spans, like sentences, might work better and reduce noise. However, segmenting the paragraphs into sentences is a challenge in our case because the original records, like most Classical Chinese texts,  lack punctuation marks which poses a segmentation challenge, and during our research, we haven't identified a tool that could efficiently address this challenge. We think that future research will look into better ways to combine these P-P interactions, and we'll leave this as a challenge for future study.
>
>
> # Regarding the reliance of one dataset for the experiments and future work
>
> We thank both reviewer ZgKy and zST9 for pointing out this concern, and we put our response to the valuable concern here.
>
> We sincerely acknowledge the concerns raised about limiting our experiments to the MingOfficial dataset. Our choice was primarily influenced by the unique nature of the MingOfficial dataset. Through our research, we found no other publicly available datasets that align with MingOfficial's characteristics, which underscores its singular presence in the domain.
>
> Our primary goal is to introduce a new paradigm for addressing similar questions in digital humanities and historical research, emphasizing the unified application of state-of-the-art technical tools in areas previously overlooked. We consider our present effort as an initial demonstration, a proof of concept that paves the way for more extensive research endeavors in the future.
>
> While our current approach was tested on a single dataset, it holds potential for wider applications across diverse datasets. To provide more context, our approach is well-suited for probing **historical documents** like Veritable Records of the Joseon Dynasty (朝鮮王朝實錄), Veritable Records of Qing (清實錄), and Chorography (地方誌). Beyond that, we can also delve into **contemporary archives**, especially those that highlight local intricacies and interpersonal dynamics, such as records from Taiwan's Legislature and Hong Kong's Legislative Council. We believe that our current work is just the beginning. It lays the groundwork for wider and more diverse research down the road.
>
> Lastly, we appreciate the reviewer's understanding regarding the intricacies of building the MingOfficial dataset. We're proactive in addressing potential biases and omissions in such a domain-specific collection. Collaborating with domain experts has been crucial in ensuring our dataset's comprehensiveness and minimizing any bias.
>
> # Regarding the typos and presentational issues:
>
> We sincerely thank the reviewer for raising these issues. We will revise the manuscript to correct presentational issues, including typos, grammar errors, missing references and the incomplete line at the end of the limitation section.

---

### Meta-Review · Area_Chair_5Wfy · 2023-09-12

**Recommendation:** 3

**Metareview:**

This paper’s main novel contribution is a new dataset of structured career records and textual historical records for around 10K career officials from the Ming dynasty. They also provide annotations for “civilian officials with military power.”

One of the challenges of a paper like this is they introduce a novel dataset, novel task (predicting “civilian officials with military power”), and a new method for the new task and dataset (specifically modeling both the structured and unstructured text using graph neural networks). Several reviewers seemed to struggle to unpack these three contributions by noting things like “the experiments were conducted on only one dataset” (Reviewer ZgKy) and “the proposed method appears to be primarily built upon existing research, and it lacks sufficient innovation” (Reviewer zST9). I think these critiques would have been mitigated if the authors had  framed this paper as primarily being a dataset contribution.

If the paper is primarily a dataset contribution paper, the dataset seems novel and important given the bias towards English-only datasets (this is in classical Chinese). As reviewer qxZu hinted at in their comment “Is the base language model used to encode historical text actually trained on ancient mandarin?”, this dataset could potentially be an valuable source for training or evaluating of models for classical Chinese.

However, I would extend the concern reviewer qxZU had “Why is military power selected as a special attribute for representation learning?” The predictive task seems interesting, although a little esoteric.

Overall, the dataset contribution is sound and exciting, even if the proposed task (predicting “civilian officials with military power”) and the methods approach with graph neural networks are a bit less exciting.

---

### Decision · Program_Chairs · 2023-10-07

**Decision:**

Accept-Main

**Comment:**

This paper’s main novel contribution is a new dataset of structured career records and textual historical records for around 10K career officials from the Ming dynasty. They also provide annotations for “civilian officials with military power.”

One of the challenges of a paper like this is they introduce a novel dataset, novel task (predicting “civilian officials with military power”), and a new method for the new task and dataset (specifically modeling both the structured and unstructured text using graph neural networks). Several reviewers seemed to struggle to unpack these three contributions by noting things like “the experiments were conducted on only one dataset” (Reviewer ZgKy) and “the proposed method appears to be primarily built upon existing research, and it lacks sufficient innovation” (Reviewer zST9). I think these critiques would have been mitigated if the authors had  framed this paper as primarily being a dataset contribution.

If the paper is primarily a dataset contribution paper, the dataset seems novel and important given the bias towards English-only datasets (this is in classical Chinese). As reviewer qxZu hinted at in their comment “Is the base language model used to encode historical text actually trained on ancient mandarin?”, this dataset could potentially be an valuable source for training or evaluating of models for classical Chinese.

However, I would extend the concern reviewer qxZU had “Why is military power selected as a special attribute for representation learning?” The predictive task seems interesting, although a little esoteric.

Overall, the dataset contribution is sound and exciting, even if the proposed task (predicting “civilian officials with military power”) and the methods approach with graph neural networks are a bit less exciting.